# Intensity Paradox—Low-Fit People Are Physically Most Active in Terms of Their Fitness

**DOI:** 10.3390/s21062063

**Published:** 2021-03-15

**Authors:** Henri Vähä-Ypyä, Harri Sievänen, Pauliina Husu, Kari Tokola, Tommi Vasankari

**Affiliations:** 1UKK-Institute, 33500 Tampere, Finland; henri.vaha-ypya@ukkinstituutti.fi (H.V.-Y.); pauliina.husu@ukkinstituutti.fi (P.H.); kari.tokola@ukkinstituutti.fi (K.T.); tommi.vasankari@ukkinstituutti.fi (T.V.); 2Faculty of Medicine and Health Technology, Tampere University, 33014 Tampere, Finland

**Keywords:** accelerometers, physical activity intensity, data processing, population sample

## Abstract

Depending on their cardiorespiratory fitness (CRF), people may perceive the exertion of incident physical activity (PA) differently. Therefore, the use of relative intensity thresholds based on individual fitness have been proposed to evaluate the accumulation of PA at different intensity levels. A subsample of the FinFit2017-study, 1952 adults (803 men and 1149 women) aged 20–69 years, participated in this study. Their maximal oxygen uptake (VO_2_max) was predicted with a 6 min walk test, and they were instructed to wear a triaxial hip-worn accelerometer for one week. The participants were divided into CRF tertiles by five age groups and sex. Raw acceleration data were analyzed with the mean amplitude deviation method in 6 s epochs. Additionally, the data were smoothed with 1 min and 6 min exponential moving averages. The absolute intensity threshold for moderate activity was 3.0 metabolic equivalent (MET) and for vigorous 6.0 MET. Correspondingly, the relative thresholds were 40% and 60% of the oxygen uptake reserve. Participants in the lowest CRF tertile were the most active with relative thresholds, and participants in the highest CRF tertile were the most active with absolute thresholds. High-fit people easily reached the absolute thresholds, while people in the lowest CRF tertile had to utilize most of their aerobic capacity on a daily basis simply to keep up with their daily chores or peers.

## 1. Introduction

Physical activity (PA) is defined as any bodily movement produced by skeletal muscles that results in energy expenditure [1]. Acute responses to PA depend on the intensity and duration of PA [2], while all these responses have different kinetics. Heart rate and oxygen consumption can reach steady state in a few minutes [3], but endocrine and metabolic changes have longer response times [4,5,6]. When PA is assessed with an accelerometer, the epoch length has a significant effect on the estimation of PA time. However, it is not known which epoch length produces the most valid estimate [7].

Positive effects of PA on health and well-being are associated with the intensity, duration, and frequency of physical effort [8,9]. Regular PA offers a safe and low-cost medicine for several health problems, and importantly, it provides an effective means to prevent these problems and related disability. However, depending on cardiorespiratory fitness (CRF), people may perceive the physical exertion or intensity of incident PA in different ways. Apparently, those with low fitness due to any reason are likely to perceive PA as being more strenuous than can be inferred from energy expenditure considered moderate in absolute terms, for example. Health benefits associated with regular PA are not only related to the total energy expenditure but also to the intensity levels at which PA is performed [10]. Therefore, it would be desirable to know both the actual intensity of PA that a person performs and the physical fitness of the given person.

The PA intensity can be expressed in absolute or relative terms [11]. Typically, the absolute thresholds for PA are defined in multiples of metabolic equivalent of an effort (MET; 1 MET = 3.5 mL(O_2_)/kg/min). The intensity varies between 3.0 and 5.9 METs for moderate PA (MPA) and is more than 6 METs for vigorous PA (VPA). The relative thresholds, in turn, are based on the oxygen uptake reserve (VO_2_R), which denotes the reserve between the resting and maximum oxygen uptake level. The recommended thresholds are 40% and 60% of VO_2_R for MPA and VPA, respectively [2]. Using these individual thresholds, the individual fitness level can be taken into account. Obviously, the same absolute intensity can be light for someone with high maximum oxygen uptake (VO_2_max) but vigorous for someone else with low VO_2_max. Therefore, the use of individual levels has been recently recommended when it is feasible [12]. In large-scale population studies, direct measurement of each individual’s VO_2_max is likely too laborious and costly [13,14,15] whereas an indirect assessment of VO_2_max may offer a feasible solution.

In the present study, VO_2_max was predicted by a 6 min walk test (6MWT), which is a well-documented method for assessing cardiorespiratory fitness (CRF) in diverse patient groups [16,17,18]. Recently, the 6MWT has been shown to be a cost-effective and valid test to assess and predict CRF among healthy adults [19]. The present study evaluated the total accumulated times of MPA and VPA using both absolute MET-based and individual fitness-based thresholds for assessing PA intensity in a large population-based sample of adults.

## 2. Materials and Methods

This study is based on a subsample of the population-based FinFit2017 study [20]. The subsample comprised of 1952 participants (803 men and 1149 women), aged 20–69 years, who completed 6MWT and wore an accelerometer 24 h per day at least for four days during a one-week data collection period. The coordinating ethics committee of The Regional Ethics Committee of the Expert Responsibility Area of Tampere University Hospital gave ethical approval for the study (R17030).

During the health examination of the study, participants’ height and weight were measured, and they performed a 6MWT. In short, participants were asked to walk back and forth along a 15 m walking track as fast as possible for six minutes. Their heart rate was recorded with a heart rate monitor (Polar M61, Polar Electro, Kempele, Finland). For men, the VO_2_max (in METs) was predicted from the walking distance in six minutes, age, body mass index (BMI), heart rate at the end of the test, and height. For women, the prediction was based on the walking distance in six minutes, body mass, and age. The accuracy of prediction was about 1 MET [19]. Based on the VO_2_max results, the participants were divided into CRF tertiles. The division was done separately for men and women in five age groups: 20–29 years, 30–39 years, 40–49 years, 50–59 years, and 60–69 years. The participant’s BMI was calculated from the measured height and weight. The BMI was categorized as normal weight (less than 25 kg/m^2^), overweight (25–30 kg/m^2^), and obese (over 30 kg/m^2^).

At the health examination, the participants received also a triaxial accelerometer (UKK RM42, UKK Terveyspalvelut Oy, Tampere, Finland) to be used for seven consecutive days all the time except during showers and other water activities. The accelerometer was attached to a flexible belt, and the participants were instructed to wear the belt so that the accelerometer was on the right side of their hip during waking hours and on the non-dominant wrist during time in bed for sleep. The acceleration signal was collected at 100 Hz sampling frequency, ±16 *g* acceleration range, and 0.004 *g* resolution. After the measurement week, the raw accelerometer data were stored on a hard disk for further offline analysis.

The accelerometer was initialized so that it started collecting data if the absolute value of the difference between reference value and the incident acceleration exceeded 187.5 m*g* in any axis and when, within the next five seconds, the difference exceeded 500 m*g* in any axis; if not, the accelerometer returned to quiescent state [21]. Whenever the raw acceleration in any axis exceeded the previous limits, the reference values were updated with the incident acceleration values. If continuous quiescent time was longer than 120 min [22], then this period was considered nonuse time.

The accelerometer data were analyzed in 6 s epochs. For each epoch, mean amplitude deviation (MAD) [23,24] and MAD*xyz* [25] values were calculated using Equations (1) and (2). In the equations, *r_i_* is the magnitude of the incident acceleration vectors *x_i_*_,_
*y_i_*, and *z_i_*; *N* is number of samples in the epoch; and *R_ave_*, *X_ave_*, *Y_ave_*, and *Z_ave_* are the mean acceleration values of the epoch.
(1)$$\mathrm{MAD}=\;\frac{1}{N}\overset{N}{\underset{i=1}{\sum }}\left|r_{i}-R_{ave}\right|$$
(2)$${\mathrm{MAD}}_{xyz}=\frac{1}{N}\sqrt{{\left(\overset{N}{\underset{i=1}{\sum }}\left|x_{i}-X_{ave}\right|\right)}^{2}+{\left(\overset{N}{\underset{i=1}{\sum }}\left|y_{i}-Y_{ave}\right|\right)}^{2}+{\left(\overset{N}{\underset{i=1}{\sum }}\left|z_{i}-Z_{ave}\right|\right)}^{2}}$$

These values were further converted to MET values depending on the instant epoch-wise MAD value using one of the following equations:(3)$$\mathrm{MET}=1.0000+0.0223\cdot \mathrm{MAD}\;\;\;\;\;\;\;\;\;\;\left(\mathrm{MAD}<91.5\;mg\right)$$
(4)MET=2.1488+0.0093 ·MAD          (91.5≤MAD<500 mg)
(5)$$\mathrm{MET}=0.4027+0.0107\cdot \mathrm{MAD}xyz\;\;\;\;\;\;\;\;\;\;\left(\mathrm{MAD}\geq 500\;mg\right)$$

Equation (3) was used if the MAD value was less than 91.5 m*g* (m*g* denotes milligravity); Equation (4) was used if MAD values between 91.5 m*g* and less than 500 m*g*; and otherwise, the Equation (5) was used. The accuracy of estimation is about 1.2 MET for bipedal locomotion over a wide range of speed [24].

The 6 s epoch MET values were further smoothed by calculating the exponential moving average (EMA) MET value for 1 and 6 min for each epoch time point (*j*) using Equation (6). In the equation, MET*_EMA_* is the smoothed MET value, MET_6s_ is the 6 s epoch MET value, and constant *N* is 10 for 1 min EMA and 60 for 6 min EMA.
(6)$${\mathrm{MET}}_{EMA}\left(j\right)={\mathrm{MET}}_{EMA}\left(j-1\right)+\frac{2}{\left(N+1\right)}\left({\mathrm{MET}}_{6s}\left(j\right)-{\mathrm{MET}}_{EMA}\left(j-1\right)\right)$$

Presumably, the 6 s epochs capture all bodily movements produced by skeletal muscles, the 1 min EMA simulates the heart rate and oxygen consumption responses to activity, and the 6 min EMA simulates the endocrine and metabolic changes.

The epoch-wise activity was classified into combined moderate-to-vigorous PA (MVPA) and VPA using both absolute and individual thresholds. The absolute thresholds were 3.0 MET and 6.0 MET, and the individual fitness-based thresholds were 40% and 60% of the VO_2_R, respectively. The classification of activity into different intensity ranges was done using three different methods by using the data based on 6 s epochs, and 1 and 6 min EMA MET values.

Accumulated MVPA and VPA times from 6 s epoch, and 1 and 6 min EMA data were calculated for the bout lengths of at least 0.1, 0.5, 1.0, 3.0, 5.0, 10.0, 15.0, 20.0, 30.0, and 60.0 min.

### Statistical Analysis

The differences between accumulated PA times in CRF tertiles were tested separately for men and women with the multiple regression analysis adjusted for age group. Natural logarithm transformations were done for all accumulated PA times to improve the normality of model residuals. One tenth of a second was added for each accumulated PA time before transformation to avoid zero values. Additionally, the mean PA times were calculated for all age groups. Partial Spearman correlations controlled for age, age^2^, and sex were calculated to quantify the degree of association between different bout lengths and VO_2_max. All statistical analyses were conducted using IBM SPSS statistics software (IBM SPSS Statistics for Windows, Version 27.0. Armonk, NY: IBM Corp., New York, NY, USA).

## 3. Results

Broken down by sex and age group, Table 1 and Table 2 show VO_2_max ranges in CRF tertiles and the corresponding proportions of participants in the three BMI groups in the given CRF tertile. Men had higher mean VO_2_max than women in all age groups and all CRF tertiles.

With the absolute 3.0 MET and 6.0 MET thresholds, participants in the high CRF tertile accumulated the most MVPA and VPA whereas participants in the low CRF tertile accumulated the least MVPA and VPA (Figure 1). This was true for both sexes with original 6 s epoch values as well as with smoothed 1 min EMA and 6 min EMA values and virtually all bout lengths. The smoothing of epoch-wise data decreased the total accumulated MVPA and VPA time, but the difference between high and low CRF tertiles in total accumulated MVPA time was at the same level with 6 s epochs and smoothed values. The high CRF tertile accumulated on average 16.4 min more MVPA time than the low CRF tertile with 6 s epoch. The corresponding difference was 15.7 min with 1 min EMA and 15.4 min with 6 min EMA. More detailed information can be found in the Appendix A.

With the individual fitness-based thresholds, the results became more complex. Participants in the low CRF tertile accumulated more total MVPA and VPA time than the ones in the high CRF tertile, both with original 6 s epoch values as well as with smoothed 1 min EMA and 6 min EMA values (Figure 2). The biggest difference in accumulated time was observed for short, less than 3 min bouts. The total accumulated MVPA and VPA times decreased with the length of data smoothing. More detailed information can be found in the Appendix A.

The relationships between the VO_2_max and total accumulated MVPA and VPA times were virtually inversed when absolute and relative intensity thresholds were employed, except for the longest bouts (Figure 3). With absolute thresholds, all correlations were statistically significantly positive (*p* < 0.05). With relative thresholds, in turn, the correlations were negative for short, less than 20 min bouts. Significantly positive correlations could be achieved only with long, over 60 min MVPA or 30 min VPA bouts. The lowest correlations (r = −0.43) were achieved with the total accumulated time from MVPA bouts at, at least, 0.1 min using 6 s epochs, and the relative threshold and the highest correlation (r = 0.38) were achieved with the total accumulated time from VPA bouts at, at least, 0.1 min using 6 s epochs and the absolute threshold.

## 4. Discussion

The main finding of the study was that low-fit persons turned out to be paradoxically the most active when individual fitness-based intensity thresholds were used for analysis. With absolute MET-based thresholds, high-fit people were the most active, as one could expect. This means that people with low VO_2_max have difficulties reaching absolute thresholds for moderate and vigorous PA, but individual relative thresholds are easily reached even during everyday habitual activities and chores. Correspondingly, it has been stated that absolute thresholds favor participants with higher fitness and relative thresholds participants with lower fitness in reaching PA recommendations [12,26,27]. The ultimate question is which one, if either, is a more appropriate viewpoint.

VO_2_max can be improved with moderate and vigorous intensity exercises, which lead to increased oxygen delivery to the working muscles and higher oxygen extraction in the muscles [28]. The optimum intensity needed for sufficient stimulus may vary depending on the individual fitness level. A more trained person seems to require higher intensities for improvements. The lowest intensity that may increase VO_2_max is 45% of VO_2_max [2], but well-trained athletes require over 95% intensities of VO_2_max to gain improvements in their CRF [2]. However, the higher intensities, at least 80% of VO_2_max, provide comparable improvements in VO_2_max in a lower time per session than lower intensities [29,30,31].

Oxygen delivery to muscles depends mainly on cardiac output [32], which is a product of heart rate and stroke volume. Stroke volume in untrained persons can reach a plateau at an intensity level of 40% of VO_2_max, but in trained persons, there is no plateau and stroke volume can increase continuously with increasing intensity [33,34]. Therefore, untrained persons can have positive effects on cardiac output with lower intensities. Oxygen extraction depends on the ability of the muscles (and other tissues) to use the delivered oxygen. Long-lasting low-intensity activities recruit mainly slow twitch muscle fibers, while higher intensity activities will recruit also fast twitch muscle fibers for relatively short durations [35]. Thus, VO_2_max development requires low intensity (50–65% of VO_2_max) and long-duration activities in combination with short, high-intensity (about 90% of VO_2_max) bouts [36].

Some people respond well to exercise training, whereas others have negligible change in their VO_2_max after similar exercises. This variability between responders and non-responders has been partly attributed to heritability, which may contribute to approximately 50% of an individual’s VO_2_max trainability [37]. However, the non-responders might require an increased exercise stimulus for a favorable response [38,39]. Thus, a more individually targeted optimal approach to exercise prescription could subsequently prove to be a powerful tool in promoting population health [39].

Physical exercise creates homeostatic stress in an individual, and homeostasis is re-established by various physiological adaptive responses. The time taken to return to resting homeostasis after physical effort depends on the volume and intensity of the exercise. Additionally, the context of the PA has an effect on the outcome. Occupational PA is often monotonous without sufficient recovery time, and it can even be detrimental for health [40]. Furthermore, lifestyle factors such as psychological stress, sleep quantity and quality, and dietary intake affect the recovery time. If there is an imbalance between overall stress and recovery, some individuals may develop fatigue and even maladaptation, yielding low or negative responses to exercise [39,41]. In the light of the present results, one can speculate that people in the low CRF tertile cannot avoid exercising. They have to utilize almost all of their aerobic capacity on a daily basis simply to keep up with their daily routines and peers. It is thus likely that their homeostasis does not have time to recover between subsequent activity bouts, impeding their fitness level from improving.

In this present study, the PA data were expressed in MET values and the 6 s epoch-wise data were further smoothed with 1 min and 6 min EMA. This filtering had a consistent effect on total accumulated MVPA and VPA times. The highest total times were obtained with 6 s epoch data, and the lowest times were obtained with 6 min EMA data. The smoothing obviously removed sporadic and fragmented short time activities. On the other hand, it can also increase the total PA time if short bouts are repeated within short recovery intervals. For example, with 6 s epochs, a 90-min football match with 15 min half-time returned 51 min total VPA (6.0 MET) time in 114 bouts, with 1 min EMA 74 min total time in 37 bouts, and with 6 min EMA 90 min total time in 6 bouts with 6 min EMA. Therefore, different data smoothing can provide different estimations of the accumulated time and knowing the context of physical activity may lead to more reliable results.

The present PA guidelines for adults no longer require physical activity to occur in at least 10-min bouts. Current evidence shows that the total volume of MVPA is related to many health benefits and that bouts of a prescribed duration are not essential [42]. The correlations obtained with absolute thresholds (Figure 3) are in line with evidence that any bout length of MVPA contributes to health benefits via better cardiorespiratory fitness. The results show that the total volume of daily MVPA or VPA in absolute terms is related to higher CRF regardless of data smoothing. Obviously, higher fitness enables moving with higher intensity. With a relative intensity threshold based on individual fitness, there is a positive but weak correlation between the total volume of PA and CRF, but only with long bouts of physical activity. While this finding may indicate that people with high fitness can sustain a high fraction of their VO_2_max for a longer time, only few participants had over 30 or 60 min MVPA or VPA bouts (from 11 to 79 participants depending on the bout). It is thus possible that the finding is sporadic and due to the behavior of the data. Notwithstanding this uncertainty, people with low fitness seem to be in a vicious circle. They exercise the most in relative terms, but low fitness persists, apparently because there is no time to recover.

In the present study, 652 participants belonged to the low CRF tertile and 564 (87%) of them had BMIs higher than 25 kg/m^2^. If all these people were able to reduce their weight such that their BMI equals 25 kg/m^2^ without any change in absolute VO_2_max (liters of O_2_ per minute), 92 (14%) of them would be classified in the high CRF tertile and 269 would be classified into the medium CRF tertile. Therefore, proper weight management could improve many participant’s CRF and could reduce the continuous physical strain from their daily activities. For the other participants in the low CRF tertile, there is a need to identify an individually optimal way to improve CRF and to be physically active. Many low-fit persons suffer from chronic diseases, such as diabetes and cardiovascular diseases, and are on medication, which all can reduce the possibilities to engage in PA.

One must also remember that PA measurement was conducted as only a one-week snapshot of the participants’ life. Previous PA training history, their intensity level, or diseases may have a significant effect on the VO_2_max [43], but history could not be considered in the present study. It would be interesting to investigate the impact of these factors on the observed associations in further studies. In addition, the present analysis was based on weekly mean values, and day-to-day variations in intensity levels and volume were not considered. The long recovery time between highly stressful PA sessions may allow for more favorable adaptations [44], whereas monotonous activity and low day-do-day variation can have undesired outcomes [45]. These topics pertaining to day-by-day and hour-by-hour variation in the amount and intensity of PA warrant further studies as well.

The use of CRF-based relative cut-points inverted common results, and the low CRF tertile became ostensibly the most active group. This paradox undoubtedly warrants some elaboration. Obviously, it is easier for high-fit, normal-weight, or young persons to reach specified absolute MVPA intensity levels than is the case for low-fit, obese, or old persons. This means that the PA intensity is underestimated among the latter groups of persons while they perceive the given PA as more strenuous than could be inferred from absolute levels. Therefore, in the clinical context, the use of an individually determined relative measure of intensity would be more appropriate for individual PA counseling and exercise prescription than an absolute measure of intensity. However, in population-based and epidemiological studies, the use of relative, individual fitness-based thresholds would likely reduce comparability between studies. In exercise interventions, in turn, the use of a relative threshold might become problematic, as the participants’ increasing fitness might reduce the amount of measured MVPA time. Consequently, in population-based studies or exercise interventions, the use of absolute thresholds would yield more reasonable results, since increasing fitness and PA would return higher measured amounts of MVPA or VPA time.

## 5. Conclusions

Using the primary 6 s epoch-wise PA data and further smoothing, the data with 1 min EMA and 6 min EMA yielded substantially different estimates of the total accumulated PA time, but they can all be consistently used to identify associations between CRF and PA. Measuring the PA in absolute or relative terms may lead to paradoxical results. High-fit people are more active when absolute MET-based thresholds are used, but low-fit people turn out to be more active and apparently more strained with relative individual fitness-based thresholds. It is likely that high-fit people easily reach the absolute thresholds while people with low fitness have to utilize most of their aerobic capacity to cope with their daily routines.

## Figures and Tables

**Figure 1 sensors-21-02063-f001:**
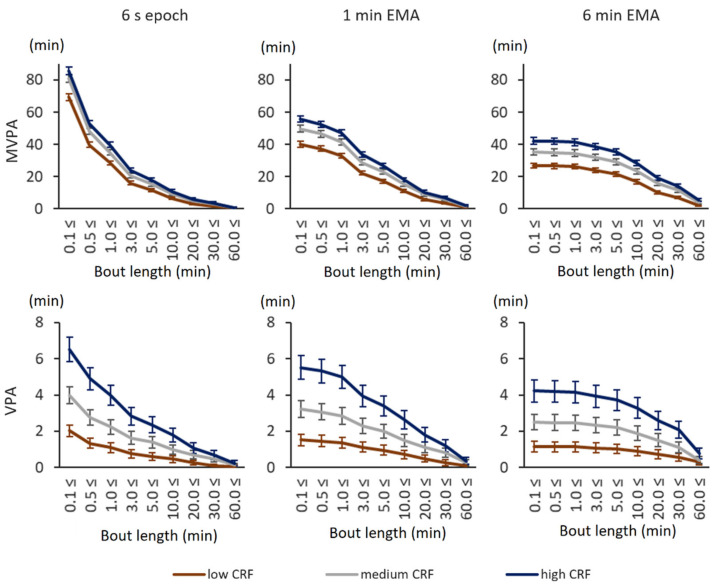
Daily moderate-to-vigorous physical activity (MVPA) and vigorous physical activity (VPA) time accumulation with absolute intensity thresholds. The *x*-axis shows the minimum accepted bout length in minutes, and the *y*-axis shows the accumulated time in minutes. The error bars denote 95% confidence intervals. The upper graphs show the accumulated MVPA time (over 3.0 metabolic equivalent (MET)), and the lower graphs show the VPA time (over 6.0 MET). The left-most graphs show the accumulated time from 6 s epoch, the middle graphs show that from 1 min exponential moving average (EMA), and the right-most graphs show that from 6 min EMA.

**Figure 2 sensors-21-02063-f002:**
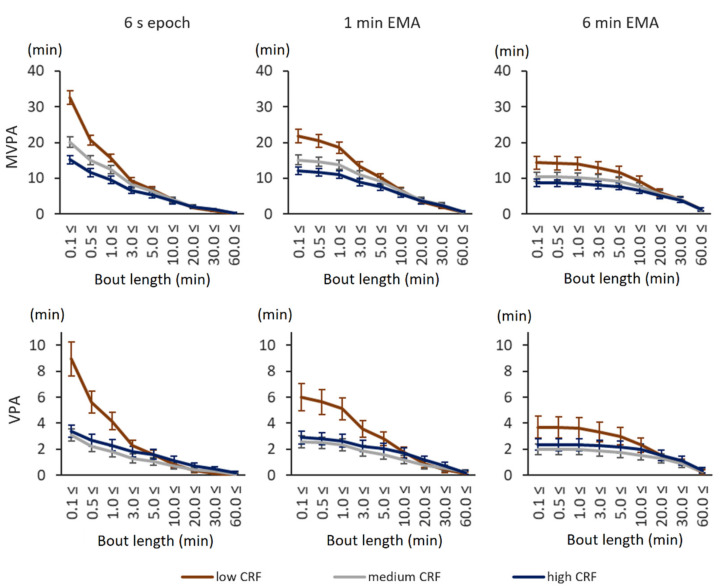
Daily MVPA and VPA time accumulation with relative fitness-based intensity thresholds. The *x*-axis shows the minimum accepted bout length, and the *y*-axis shows the accumulated time in minutes. The error bars denote 95% confidence intervals. The upper graphs show the accumulated MVPA time (over 40% of VO_2_R), and the lower graphs show the VPA time (over 60% of VO_2_R). The left-most graphs show the accumulated time from 6 s epoch, the middle graphs show that from 1 min EMA, and the right-most graphs show that from 6 min EMA.

**Figure 3 sensors-21-02063-f003:**
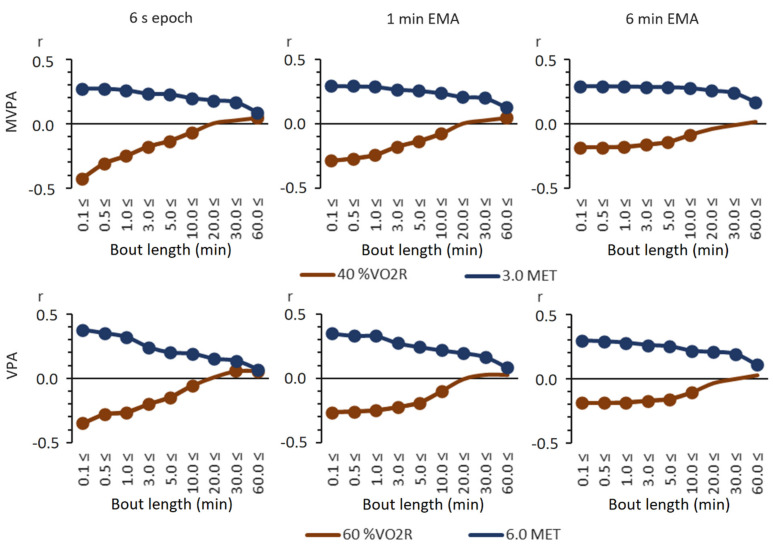
Partial Spearman correlations controlled for age, age^2^, and sex between VO_2_max and time accumulated from MVPA and VPA bouts at the time limit with absolute and relative thresholds. The *x*-axis is the time limit in minutes, and the *y*-axis is the correlation coefficient. The round marker on the line denotes a statistically significant correlation (*p* < 0.05).

**Table 1 sensors-21-02063-t001:** Number of participants in each age group and maximal oxygen uptake (VO_2_max) range (mL/kg/min) in cardiorespiratory fitness (CRF) tertiles broken down by sex and age group.

Age Group	*N*	Low CRF	Middle CRF	High CRF
Men				
20–29	55	18.0–41.6	41.6–44.2	45.2–53.4
30–39	133	27.5–38.2	38.2–42.1	42.2–53.3
40–49	160	19.7–35.3	35.5–41.0	41.0–50.2
50–59	191	15.9–32.7	32.8–37.1	37.1–46.4
60–69	264	11.2–29.1	29.2–33.6	33.6–43.8
Women				
20–29	120	20.1–35.0	35.0–38.6	38.7–48.6
30–39	185	20.3–33.4	33.4–38.1	38.1–46.3
40–49	236	17.3–32.6	32.6–37.2	37.2–47.0
50–59	267	9.9–28.6	28.6–33.7	33.8–43.0
60–69	341	14.5–27.0	27.1–31.6	31.6–41.5

**Table 2 sensors-21-02063-t002:** Proportion of participants in each BMI groups (normal weight, overweight, and obese) in CRF tertiles broken down by sex and age group.

Age Group	Low CRF			Middle CRF			High CRF		
	Normal Weight	Over-Weight	Obese	Normal Weight	Over-Weight	Obese	Normal Weight	Over-Weight	Obese
Men									
20–29	28%	61%	11%	50%	39%	11%	84%	16%	0%
30–39	30%	32%	39%	32%	66%	2%	60%	40%	0%
40–49	9%	49%	42%	32%	60%	8%	69%	30%	2%
50–59	11%	45%	44%	29%	63%	8%	45%	48%	6%
60–69	13%	47%	41%	24%	58%	18%	58%	38%	5%
Women									
20–29	38%	45%	18%	78%	23%	0%	98%	3%	0%
30–39	24%	45%	31%	72%	25%	3%	92%	6%	2%
40–49	8%	38%	54%	71%	29%	0%	94%	6%	0%
50–59	1%	39%	60%	39%	56%	4%	82%	18%	0%
60–69	9%	43%	48%	36%	55%	9%	77%	23%	0%

## Data Availability

The datasets of the current study are not publicly available due ethical reasons. Non-identifiable data are available for research purposes from the corresponding author upon reasonable request.

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
