# Peer review of "Intensity Paradox—Low-Fit People Are Physically Most Active in Terms of Their Fitness"

_sensors, 2021, doi:10.3390/s21062063_

Round 1

Reviewer 1 Report

Units should be written with a space spacing, for example, 6s => 6 s. Furthermore, chemical formulas O2 should be written according to the rules of stoichiometry and SI kg–1 => kg^(–1).
Equations 1, 2, and 3 are not logical. MET can be defined with 3 equations but put in index MAD consecration. In 3 equation mean amplitude 99 deviation has index xyz? Line 139 ''nor-mal weight'' normal has a dash. Table 1 (28% - 61% - 11%) describe the content in parentheses. The units in Figure 1 must be in parentheses (min). x-axis should be described in the figure not only in the title of the image.

Author Response

Thank you for your comments.

The units are now rewritten with a space spacing and their presentation is revised.

The chemical formula of oxygen is now written as O2 throughout the text.

The concern on the Equations 1,2, and 3 (now 3,4, and 5) is straightened up by adding two equations describing the calculation of MAD and MADxyz (new equations 1 and 2). Depending on the magnitude of the incident epoch-wise MAD value, concurrent MET is estimated using the designated equation 3, 4, or 5. Hopefully these changes clarify things in the Equations.

The line 139 typo is corrected.

Table 1: the contents in the parentheses denote the proportions of normal, overweight and obese person within the given age group. The grouping was based on the person’s BMI as described in the text. However, for the sake of clarity, we have now divided Table 1 into Tables 1 and 2. Table 1 describes now the ranges of VO2max in the age-group and sex-specific CRF tertiles, whereas table 2 describes the proportions of normal, overweight and obese participants in the age-group and sex-specific CRF tertiles. Hopefully this clarifies things.

The Figure 1 is corrected, and the units are now given in parentheses. Also, the x-axis is described in the figure as well

Reviewer 2 Report

In the current study, VO2max was predicted by 6-min walk test, which is a reliable index of cardiorespiratory fitness. The study's findings could be improved by better analysis and presentation of findings. For example, Table 1 is crowded with information that can't be easily concluded. I suggest using BMI and age as continuous variables and evaluating their relationships with CRf and VO2max more appropriately. Furthermore, the sample size is large enough to conduct subgroup analyses.

Author Response

Thank you for your comments. We quite agree the issue concerning the analysis and presentation. In fact, we have elaborated quite a bundle of different ways to illustrate this complicated topic concerning the absolute or relative fitness, age, body weight, and sex, and eventually ended up with presenting the accumulated amounts of MVPA and VPA from different bout lengths within the context of individual CRF. As the study stands now, the present data illustrates the inherent paradox that pertains to the individual CRF and its impact on apparent perceived exertion and accumulated PA. Apparently, these association may be modulated by BMI and age, but should have we used these variables as continuous variables we would not have been able to illustrate this issue as clearly as now – our primary purpose was to demonstrate the presence of this paradox. In addition, we investigated the effect of different smoothing filters (1 min and 6 min EMA filters, now described in Equation 6, on accumulation of PA from different bouts. Therefore, we prefer to stick to our original approach. Addressing multiple factors in a single study would make this already complicated paper even much more complex. Therefore, we propose further studies in the first paragraph on page 10.

For the sake of clarity, we have now divided the busy Table 1 into Tables 1 and 2. Table 1 describes now the ranges of VO2max in the age-group and sex-specific CRF tertiles, whereas table 2 describes the proportions of normal, overweight and obese participants in the age-group and sex-specific CRF tertiles. Hopefully this clarifies things.

Reviewer 3 Report

The experiments and results presented in this paper are in line with the standard methodology. The large number of volunteers involved in the experimental part is impressive and represents one of the positive points of this approach. The conclusions of this paper complete the previous results obtained by the authors in the approached field.

Some details regarding the data acquisition, storage, and preprocessing system would have been welcome. It would also have been of interest to split the users, taking into account their previous personal physical training.

Author Response

Thank you for your comments.

The data acquisition parameters were described by sampling frequency, resolution and dynamic range on lines 89-90 of the initial submission.

As requested, we have now provided the equations for 1 min and 6 min EMA filters (equations 6 and 7).

We agree that it would be interesting to investigate the influence of different factors on the fitness and its impact on accumulation of PA, such as previous physical training. We brought up this relevant point in the first paragraph on page 10.